# Towards Automatic Expressive Pipa Music Transcription Using Morphological Analysis of Photoelectric Signals

**DOI:** 10.3390/s25051361

**Published:** 2025-02-23

**Authors:** Yuancheng Wang, Xuanzhe Li, Yunxiao Zhang, Qiao Wang

**Affiliations:** 1School of Information Science and Engineering, Southeast University, Nanjing 211102, China; wangyuancheng@seu.edu.cn; 2School of Artificial Intelligence, Southeast University, Nanjing 211102, China; 213213686@seu.edu.cn (X.L.); 213213840@seu.edu.cn (Y.Z.)

**Keywords:** automatic music transcription (AMT), pipa, playing techniques, amplitude modulation-frequency modulation (AM-FM), photoelectric signal, morphological analysis

## Abstract

The musical signal produced by plucked instruments often exhibits non-stationarity due to variations in the pitch and amplitude, making pitch estimation a challenge. In this paper, we assess different transcription processes and algorithms applied to signals captured by optical sensors mounted on a pipa—a traditional Chinese plucked instrument—played using a range of techniques. The captured signal demonstrates a distinctive arched feature during plucking. This facilitates onset detection to avoid the impact of the spurious energy peaks within vibration areas that arise from pitch-shift playing techniques. Subsequently, we developed a novel time–frequency feature, known as continuous time-period mapping (CTPM), which contains pitch curves. The proposed process can also be applied to playing techniques that mix pitch shifts and tremolo. When evaluated on four renowned pipa music pieces of varying difficulty levels, our fully time-domain-based onset detectors outperformed four short-time methods, particularly during tremolo. Our zero-crossing-based pitch estimator achieved a performance comparable to short-time methods with a far better computational efficiency, demonstrating its suitability for use in a lightweight algorithm in future work.

## 1. Introduction

Automatic music transcription is a fundamental signal processing task in music information retrieval (MIR). It aims to convert an acoustic signal into some form of music notation [1]. The pipa is a traditional Chinese plucked string instrument with origins dating back to around the 2nd century BC. Its earliest form may have originated from instruments in Central Asia and regions to the west, such as Persia or India. These instruments made their way into China through trade and cultural exchanges along the Silk Road, evolving and localizing over time to become the pipa that we are familiar with today. The modern pipa is played with fake celluloid nails worn on all the fingers of the right hand and has four strings, six ledges, and 24 frets. As shown in [2], there are more than 60 playing techniques for the pipa, including pitch-shift techniques (such as vibrato, sliding, and bending) and tremolo, which is achieved by a series of rapid plucking movements. In this paper, we preliminarily study converting the the pipa string’s displacement signal into music notation using photoelectric pickups.

The performers manipulate the properties of the pipa’s sound beyond the pitch and duration determined by composers to enrich musical expressiveness and human perception [3]. These manipulations include the pipa’s fundamental frequency (F0, equivalent to pitch and reciprocal of period), timing, amplitude, and timbre (harmonic spectrum). In a typical performance, these auditory parameters do not vary. In an expressive performance, various playing techniques are used, creating intensive strength and pitch variations. The latter techniques are addressed in this paper. In previous work, various monophonic transcription tools, such as Tony [4] that employ short-time approaches using local stationarity were assessed. It was shown that they did not perform well on the guitar [5], on Chinese bowed instruments from the huqin family [6], or with string and vocal quartets [7,8].

Automatic pipa transcription (APT) is an automatic music transcription (AMT) task dedicated to the pipa. It aims to build a system capable of handling the varied playing techniques and that can extract features such as the string number, F0 (fundamental frequency), boundaries (attack+transient+offset), and notes played (attack+offset+integer MIDI number, in which the MIDI number *n* can be converted into a frequency in Hz using $440\times 2^{(n-69)/12}$ [9]). The challenges in this context are listed below:The pipa is a polyphonic music instrument that can be played by plucking multiple strings simultaneously;Pitch-shift techniques physically introduce time-varying pitch and amplitude fluctuations. The former require a high frequency-domain resolution and the latter often produce spurious energy peaks that can mislead energy-based onset detectors;Tremolo produces rapid amplitude fluctuations with a maximal tremolo rate (tremolo plucking speed) of up to 20 Hz, which requires a high time-domain resolution;A mixture of these playing techniques can occur simultaneously, which makes the window size used in short-time approaches difficult to select;In addition to a higher accuracy and resolution, a faster algorithm running speed is also important to improve the user experience.

To obtain plucked instruments’ polyphonic music features, Xi [10] and Wang [11] mounted electromagnetic pickups on a guitar and photoelectric pickups on a pipa. The polyphonic signals could, thus, be de-mixed into monophonic signals, and the multipitch estimation task could be converted into a single pitch estimation task for each of the different strings. These two works provide a process with four steps, i.e., source separation to reduce mutual resonance among strings, boundary detection, pitch estimation, and note segmentation; however, all steps are implemented using short-time algorithms and neither of these studies investigates their transcription performance in terms of boundary and pitch.

From a morphological perspective, the electromagnetic signal has a waveform similar to that of an audio wave. Figure 1A shows a clip of an electromagnetic signal with two sliding tones that create two spurious energy peaks at around 0.4 and 1.6 s (indicated by cyan lines). In Figure 1B, a segment of *Love of the Wei River*, which contains a mixture of tremolo and pitch-shift techniques, is taken as an example. The tremolo, vibrato, and sliding techniques used imitate the Qin Opera vocal style by controlling the intensity and pitch of the performance. Since the photoelectric pickup presented in [11] directly captures the string displacement, and plucks severely bend strings for a very short time, its photoelectric signal has an arched form during plucking, as shown in Figure 1B and Figure 2. The two ends of an arched form, known as the attack and transient, represent the instances where the fake nail touches the string and the string begins to vibrate, respectively. The arched form is not apparent in the signal from microphones or electromagnetic pickups as a result of their suppressed low-frequency response; however, it could circumvent the amplitude fluctuation introduced by pitch-shift techniques such as vibrato and sliding during signal vibration. After boundary detection, we can focus on non-stationary vibration signal pitch estimation within each clip. Figure 1C presents the CPTM of the waveform in Figure 1B; the red points denoting the pitch curves are included in CPTM points. The gaps correspond to the attack–transient intervals, during which the signals are arched and do not exhibit periodicity. Therefore, after note segmentation, interpolation was used to fill the pitch gaps within the tremolo notes, as shown in Figure 1D.

More concretely, a clip of a periodic signal with amplitude modulation (AM) and frequency (period) modulation (FM) [13] can be formulated as follows:(1)S(t)=A(t)S(t−T(t))
where A(t) and T(t) denote the time-varying amplitude (envelope) and period, respectively.

The zero-crossings (ZCs) and extrema are two important concepts in dealing with the AM-FM periodic signal used in this paper. The continuous-valued time instants of zero-crossing S(t)=0 and extrema (local maxima or minima, S(t)=max/min) can be achieved using linear interpolation of the adjacent non-positive and positive sample points and parabolic interpolation of three adjacent sample points around the peak or trough of the digital signal. The pitch period achieved using the time interval of adjacent pitch markers (PMs) (constructed by zero-crossings or extrema) naturally disentangles from the amplitude envelopes and does not require window size selection to frame the signal. However, since the waveform may have multiple oscillations in a period, choosing a series of zero-crossings or extrema corresponding to the pitch markers and detecting the arched form become two core problems in this paper (see Figure 2). The above-mentioned continuous time-period mapping (CTPM) technique proposed in this paper can be used to choose the pitch markers over time.

The time intervals in our methods are mainly based on the pipa register, which ranges from 110 to 1318.5 Hz, with the raw signal low-frequency information (<110 Hz) being used for onset detection, and the high-frequency information (>1318.5 Hz) being neglected in all proposed methods. The arched form benefits from the photoelectric sensors’ full-band response and can be seen as a novel low-frequency feature. Thus, it is further explored in this paper.

The main contributions of this paper are as follows:Compared to previous work, our methods provide more precise pipa boundaries by applying new boundary detectors using zero-crossing or extrema time intervals;A novel time–frequency feature, i.e., continuous time-period mapping (CTPM), is proposed to track the pitch curves without the need for waveform preprocessing. It is constructed using zero-crossing or extrema time intervals;Note segmentation for tremolo and pitch curve post-processing within tremolo notes are proposed to help in the future analysis of tremolo pitch-shift techniques.

The remainder of this paper is organized as follows. In Section 2, we briefly review the related work regarding commonly used onset detectors, pitch estimators, and transcription processes. The proposed fully time-domain-based methods for boundary and pitch are outlined in Section 3. The experimental results and related discussion are given in Section 4 and Section 5. Section 6 concludes the paper and describes promising future work.

## 2. Related Works

Monophonic transcription consists of unvoiced/voice detection (UVD) (also known as voice activity detection (VAD) [14]) and pitch estimation steps. For a new instrument, pitch estimation is generally implemented by completely explainable unsupervised algorithms that can be categorized into parametric, non-parametric, and time-domain approaches. The former two hypothesize the local stationarity of the short-time signal. For example, a typical parametric approach, non-linear least square (NLS [15]), decomposes the signal into a linear combination of Fourier bases adhering to the properties of the stationary waves and harmonic structure. Shi proposed the Bayesian non-linear least square (BNLS [16]) approach by modeling the order prior and temporal continuity based on pitch and voice activity. Most non-parametric approaches, such as the autocorrelation function (ACF [17]), capture the period of speech and music signals and neglect the timbral characteristic represented by the amplitude of each harmonic component (harmonic spectrum) in NLS. A state-of-the-art probabilistic YIN (pYIN) [12] addresses the impact of envelope fluctuation using an improved version of ACF—the cumulative mean normalized difference function (CMNDF [18], $d\left(t\right)t/\sum_{j=1}^{t}d\left(j\right)$ where $d\left(t\right)=\sum_{j=1}^{W}{\left(s_{j}-s_{j+t}\right)}^{2}$, with *s* being the input signal, *t* the sample distance, and *W* the window size). In addition, it also addresses the temporal continuity on pitch and voice activity. However, pYIN and BNLS voice activity detection (VAD) based on the frame-wise signal reconstruction quality represented by the voicing probability cannot achieve a note’s accuracy boundaries.

Since onsets and offsets are often characterized by sudden changes in amplitude, boundary detection provides another way to limit note ranges with higher resolution [4]. The frame-wise root mean square (RMS) envelope of the high-pass signal, referred to as log-energy in the following part of this paper, has been used for intensive plucking, such as tremolo [19]. SpecFlux [20], SuperFlux [21], and ComplexFlux [22] are another three common algorithms. The first has also been applied to the guitar [10]. Since vibrato and tremolo produce spectral changes in the frequency and amplitude and lead to a significant spectral difference, X(t,f)−X(t+1, f) is used in SpecFlux between adjacent frames, with X(t,f) being the amplitude spectrum of the input signal in frame *t* and bin *f*. In order to only obtain the note onset, the latter two replace the spectral difference, with maximum filtering acting on the three bins along the frequency domain X(t,f)−maxX(t+1, f−1:f+1) and the phase difference ϕ(t,f)−ϕ(t+1, f), with ϕ denoting the 2π-unwrapped phase of X(t,f).

Time-domain pitch estimators aim to capture zero-crossing (ZC) or extrema sequence as different kinds of key feature points. Multiple local candidates within a period may lead to difficulty in selecting pitch markers. In empirical mode decomposition (EMD [23]), the intrinsic mode function (IMF) is computed by the mean curve of the upper and lower envelopes passing through the extrema. This has already been applied to endpoint detection [24] and pitch estimation [25] in speech. However, it does not ensure the periodicity of the IMF.

Hess [26] summarized the two categories of time-domain pitch estimators. The first category relies on low-pass filtering to achieve a single harmonic component in a period. However, the moderate cut-off frequency depends on the pitch for estimation, and an arbitrary filter inevitably warps the raw signal. The DIO algorithm [27] in the WORLD vocoder [28] analyzes the candidate pitch markers after low-pass filtering of different cut-off frequencies. It selects F0 if its four intervals are approximately identical, i.e., those of the two adjacent minima, maxima, non-positive-to-positive zero-crossings and positive-to-non-positive zero-crossings. This algorithm has a high running speed and performance comparable to the short-time pitch estimators at that time. The second category is based on thresholding the raw signal. Most related studies appeared before the 1990s and require a waveform with very few oscillations. In addition, cascaded approaches first estimate a pitch curve using a robust short-time detector and then capture the pitch markers with the reference pitch obtained by the short-time approach. Granular synthesis [29] and pitch-synchronous overlap add (PSOLA [30]) employ zero-crossings and extrema, respectively, to split the waveform into clips (grains), with the time-varying lengths referring to the pitch periods obtained by short-time approaches. Using these methods, the music signal with an arbitrary pitch and amplitude can be synthesized by resampling and modifying the envelope on each grain. In the following, we propose various time-domain approaches for a relatively complex signal without filtering.

Finally, we categorize the transcription processes into pitch estimation before segmentation (PES) and segmentation before pitch estimation (SPE). The original probabilistic YIN (pYIN) and Bayesian non-linear least square (BNLS) techniques attributed to the PES process determine the voice activity by the quality of the signal reconstruction after pitch estimation. Time-domain approaches such as ours use the SPE process. In this, boundary detection provides a structural prior to constraining the pitch ranges to be estimated. The pitch curve throughout time can also be retained by setting the voicing probability threshold to 1 in pYIN and BNLS. It is then assigned by the corrected boundaries in each voiced clip. Although voice activity detection (VAD) from the original pYIN and BNLS is neglected, we denote this process PES in this paper.

## 3. Methodology

After removing the string resonance using kernel additive model interference reduction (KAMIR) [11,31], the monophonic string displacement signals can be processed with the following fully time-domain-based steps to obtain the remaining features required for automatic pipa transcription (APT), i.e., the pitch, boundaries, and notes. The first three steps detect the boundaries; the last three address the pitch and notes.

Onset candidate selection: localize the attack–transient pair at two ends of the arched waveform by analyzing the time-intervals of the zero-crossings or envelope zero-crossings;Onset fine-tuning: Remove the attack–transient pair if the candidate transient does not meet the conditions of sudden change to improve the precision. Fine-tune the remaining attack–transient pairs to accurate time instants;Vibration signal extraction: determine the offset between a transient and its subsequent attack (the subsequent attack may coincide with the current offset), and extract the vibration signal clip between the transient and offset for pitch marking, which aims to select a pitch marker sequence in the vibration signal;Pitch marking: select an initial pitch marker pair in a vibration signal clip, and track all the pitch markers forward and backward via continuous time-period mapping (CTPM);Note segmentation: Eliminate the attack–transient pairs within tremolo notes according to the pitch continuity and transient–offset intervals;Pitch interpolation: Smooth the pitch curve within the boundaries of a note and interpolate the pitch curve clips to fill the gaps within a tremolo note.

### 3.1. Boundary Detection

Onset candidate selection: Three methods are proposed to capture the two ends of the arched waveform. In the first method, the absence of a subsequent zero-crossing within an interval of 9.09 ms corresponding to 110 Hz signifies a candidate attack–transient pair. The second method checks whether the upper or lower envelope composed of the extrema passes through the coordinate axis. The extrema can be obtained by pick-peaking with a minimal distance of 33 sample points at a 44,100 Hz sampling rate corresponding to 1318.5 Hz. Moreover, only a single extremum is maintained in each 400-sample interval to ensure one extremum per period at most. The third method, termed the extrema-with-padding method, is a modified version of the second, which fills in the extrema to ensure that the minimum and maximum alternate. This allows for the attack–transient pair with an interval shorter than 9.09 ms to be partially captured and reduces false negatives (see Figure 3A).

Onset fine-tuning: The degree of change in the amplitude is an indicator to distinguish the transients. In this step, two extrema with different signs around a transient are extracted and the change in the amplitude is measured via the width ratio of the envelopes at these two extrema to reduce false positives during silence. The correct transient is annotated in the previous extremum. In Figure 3B, the correct attack is corrected to the inflection point after the candidate attack (coarse attack) produced by the finger release. Inflection is determined by the maximum absolute value of the slope difference of two adjacent mean curve segments. The attack–transient intervals are limited to 150 ms, which also provides a threshold to remove false positives from silence areas.

Vibration signal extraction: As shown in Figure 3C, the offset of a note is defined in the moment of extremum when the upper or lower envelope passes through the axis after the transient. Therefore, given the sample-level resolution boundaries, a clip of signal within a transient–offset pair in a non-tremolo note and multiple clips in a tremolo note count in signal vibration areas for subsequent pitch estimation.

### 3.2. Pitch Estimation

The pitch estimation process is shown in Figure 4. First, we extract pitch markers (zero-crossings or extrema). At most, one pitch marker appears within 33 sample points at a 44,100 Hz sampling rate. Figure 4A depicts an example of extrema extraction. Then, continuous time-period mapping (CTPM) and its time-domain density function are calculated to select a window (see Figure 4B), which helps to extract the initial period through the period density function (see Figure 4C). The yellow lines in Figure 4D represent the extrema pair corresponding to the initial CTPM point. Starting from these points, the tracked pitch markers in both directions are indicated by the red lines in Figure 4D, which also correspond to the red CTPM pitch points shown in Figure 4B. Various details of the process are described below.

Continuous time-period mapping (CTPM): Since pitch markers are not easy to select if a few zero-crossings or extrema candidates occur within a period, we propose a novel time frequency feature to analyze the signal’s period/pitch structure: the continuous-valued time instant of the *i*th pitch marker candidate, $t_{i}$. To avoid an excessive amplitude shift and signal envelope asymmetry, we eliminate the mean curves of the upper and lower envelopes (the first IMF of empirical mode decomposition (EMD)) from the signal clip. The relevant formulas are described below:(2)$$\begin{array}{cccc} p_{i,j} & =\frac{1}{t_{i+j}-t_{i}} & \;\;\;\;\;\;\;\;\;\;\;\;\;\;\;\;\;\;\;\;\;\;\;\;T_{i,j} & =\frac{t_{i+j}+t_{i}}{2} \end{array}$$(3)$$\begin{array}{cccc} t_{i} & =\frac{1}{2p_{i,j}}-T_{i,j} & \;\;\;\;\;\;\;\;\;\;\;\;\;\;\;\;\;\;\;\;\;\;\;t_{i+j} & =\frac{1}{2P_{i,j}}+T_{i,j} \end{array}$$
where i,j≥1, the pitch period $p_{i,j}$ and its time instant $T_{i,j}$ of a point in CTPM correspond one-to-one with $t_{i}$ and $t_{i+j}$, and only the distance $t_{i+j}-t_{i}$ in pipa register counts in CTPM candidate points.

Pitch marker pair initialization: We select a window of varying length using a marginal distribution in the time domain for the extraction of the initial pitch period (see the pink interval in Figure 4B). The approximate initial instant is defined in the middle of the window. This window is located in an area with few candidate points.

Since the points in CTPM are continuously valued both in time and in period, a Gaussian kernel density estimation method [32] is used to implement the marginal distribution. Assuming the timestamps of K CTPM points within a certain range are $T_{k}$ (*k* = 1, …, *K*), the corresponding time-domain distribution can be formulated as follows:(4)$$\begin{array}{c} P\left(t\right)=\frac{1}{K10^{-2}\sqrt{2\pi }}\sum_{k=1}^{K}\exp\left(-\frac{{\left(t-T_{k}\right)}^{2}}{2\times 10^{-4}}\right) \end{array}$$
where the standard deviation of the Gaussian kernel is $10^{-2}$ s. After obtaining the P(I) distribution, the peaks serve as two ends of the window. The window length is limited by a 0.1 s maximum and the duration of the clip. Similar to Equation (Equation 4), once the window is chosen, the period-domain marginal distribution P(I) of CTPM candidate points can be obtained using Equation (Equation 5), assuming the variance of the Gaussian kernel as one sample distance and the periods of L CTPM points within the selected range $I_{l}$ (*l* = 1, …, *L*):(5)$$\begin{array}{c} P\left(I\right)=\frac{1}{L\sqrt{2\pi }}\sum_{l=1}^{L}\exp\left(-\frac{{\left(I-I_{l}\right)}^{2}}{2}\right) \end{array}$$

Inspired by non-parametric pitch estimation methods such as autocorrelation function (ACF) [17], we select the sample distance of the first peak greater than λ× the maximum with λ∈(0,1) as the initial CTPM point’s approximate period. Thus, the corresponding pitch marker pair can be achieved, and the other pitch markers can be inferred before and after this pair using Equation (Equation 6).

Pitch marker tracking: The next continuous reference interval $I_{n+1}$ is updated by the current interval $I_{n}$ and the distance $t_{i_{n}\pm j_{n}}-t_{i_{n}}$, and a momentum parameter β is used for pitch marker tracking:(6)$$I_{n\pm 1}=\beta I_{n}+\left(1-\beta \right)\left(t_{i_{n}\pm j_{n}}-t_{i_{n}}\right)$$
where $j_{n}=\arg\min_{j}\left(\right|t_{i_{n}\pm j}-\left(t_{i_{n}}\pm I_{n}\right)\left|\right)$, *n* represents the number in the pitch marker sequence, and $i_{n\pm 1}=i_{n}\pm j_{n}$.

The pitch curves estimated using the proposed methods need to be smoothed using an outlier-robust method to ensure their pitch continuity. The robust locally weighted scatterplot smooth algorithm (rLoWeSS [33]) was used in this study.

### 3.3. Note Segmentation and Post-Processing of the Pitch Curves in Tremolo Notes

For note segmentation, if the offset of the previous clip coincides with the attack of the current clip, the current clip can be merged into the previous one when the duration of its current transient–offset pair is shorter than a threshold of τ seconds and the difference between the last pitch value of the previous clip and the first pitch value of the current clip is less than 1 semitone. Note that the durations of the first and last clips in a tremolo note are allowed to exceed τ seconds.

Finally, in future studies of pitch-shift playing techniques, the pitch gaps within the tremolo notes will need to be filled using interpolation. In practice, cubic interpolation was utilized for this step in this study. Figure 5 shows the final automatic pipa transcription (APT) features from an expert of *Love of the Wei River*, including polyphonic notes, tremolo, vibrato, bending, and sliding techniques, where the pitch gaps in the first attack–transient intervals in the tremolo notes are maintained as normal notes.

## 4. Experiments

### 4.1. Datasets, Evaluation Metrics, and Parameter Setting

In this section, we examine the enhanced PipaSet’s transcription performance using four excerpts of famous traditional Chinese pieces from the pipa grade examination [2]. Two low-level pieces, *Jasmine Flower* (LvL. 1) and *Nanni Bay* (LvL. 2), and two high-level pieces, *Love of the Wei River* (LvL. 8, newly added) and *Ambush from Ten sides* (LvL. 8), were performed naturally. High-level pieces are longer, have greater polyphony, and use more intricate playing techniques. Only vibrato and sliding occur in *Jasmine Flower* and the tremolo rates in *Nanni Bay* and *Love of the Wei River* are not particularly high, while vibrato, sliding, tremolo, and even a combination of these are widely present in *Love of the Wei River* and *Ambush from Ten Sides*. In addition, the latter contains a large quantity of intensive tremolo plucks with attack–transient intervals shorter than 9.09 ms. The other playing techniques, such as harmonic, twisting, and percussion, account for a very small proportion. Four photoelectric signal channels totaling 984 s were recorded with a standard 44,100 Hz sampling rate and 24 bit depth.

The boundaries were re-annotated following the sample-level framework as outlined in Section 3.1. Since attacks and transients are only distinguished by the methods proposed in this paper, we merged them for evaluation with the other algorithms. Moreover, only the initial attack–transient pair of the tremolo note was considered in the note onset evaluation, regardless of the ground-truth or the estimated ones, as all playing techniques were excluded in a note segment. The onset metrics precision (P=TP/(TP+FP)), recall (R=TP/(TP+FN)), F-score (F = the geometric average of *P* and *R*) were implemented using mir_eval [34], with TP, FN, and FP denoting the numbers of true positive, false negative, and false positives. An estimated onset was a true positive if it was located within 50 ms (±25 ms) of the correct one. To avoid the impact of note numbers performed on different strings, the final F/P/R was the average weighted by the pluck numbers of each string.

Although the tremolo technique has been barely studied, the plucks in the tremolo note can be seen as equivalent to the note onsets and used to identify if a note is characterized by tremolo. First, we examined the location of tremolo plucks using the same onset metric and denoted it tremolo pluck detection (TPD). Second, a note was identified as a tremolo if there were at least two plucks or four peaks present. Tremolo note recognition (TNR), which represents the number of correct notes over the number of annotated notes, is proposed as another tremolo metric.

The resolution of the correct note onset and tremolo plucks was measured using mean absolute error (MAE). The intersection over union ($IoU\left(G,E\right)=\frac{G\cup E}{G\cap E}$ with the ground-truth G and the estimated intervals E) was used to evaluate voice activity detection (VAD) in existing pitch algorithms and the proposed boundary detectors.

For pitch estimation, we only assessed the integer MIDI numbers that were converted from all median pitches (in Hz) in each note due to the inaccessibility of a perfect reference F0 [4]. Based on the same vibration signal areas to reduce the variable, the pitch estimation accuracy indicates the number of notes with the correct estimated pitch over the number of annotated notes.

Through statistical analysis and parameter scanning, we obtained the envelope width ratio (EWR) > 0.77, sudden slope change (SSC) > 2.5, momentum β=0.8, pitch peak ratio λ=0.765, note segmentation threshold τ=0.122, and the range multiplier of rLoWeSS = 0.1.

### 4.2. Experimental Results

#### 4.2.1. Boundary Results

According to the results in Table 1, the extrema-with-padding method achieved recall improvements of 6% and 24%, respectively, as compared to those of log-energy and SpecFlux. Extrema (without padding)- and zero-crossing (ZC)-based methods exhibited the same overall F-score. This is a 19.25% and 8.25% increase compared to log-energy and SpecFlux. Although the proposed methods and log-energy had a sufficiently high recall and resolution, our methods generally outperformed log-energy in terms of F-score.

In the tremolo case shown in Table 2, the extrema-with-padding method exhibited a 14% improvement in TNR recall compared to log-energy, the extrema (without padding) method showed a 21.33% and 33.67% increase in TNR F-score compared to the SpecFlux and log-energy methods. The extrema-with-padding method exhibited a 34% improvement in recall and a 21.66% improvement in F-score compared to log-energy under the TPD metric. Our detection method outperformed the other algorithms in terms of F-score for tremolo cases on all pieces, and the extrema-with-padding method achieved the best recall and F-score in *Ambush from Ten Sides*.

Both the log-energy and the proposed algorithms correctly estimated the boundaries with a high resolution. However, most of our algorithms’ boundaries exhibited zero deviation when identified. Our methods reduced the MAE by approximately 80% compared to the first three short-time methods. Finally, the VAD metric in Table 3 validates the effectiveness of boundary detection compared to reconstruction-based VAD for pitch estimators.

#### 4.2.2. Pitch and Note Results

According to the results in Table 3, the zero-crossing (ZC)-based pitch estimator exhibited a slight performance loss compared to the probabilistic YIN (pYIN) and Bayesian non-linear least square (BNLS) methods using the pitch estimation before segmentation (PES) process. Moreover, catastrophic results were recorded for the pYIN and BNLS methods using the segmentation before pitch estimation (SPE) process. Figure 6 presents the pitch deviation distribution computed by $\frac{E-G}{\#G}$ with the estimated pitch E and the ground-truth G in units of the integer MIDI number for each note among all algorithms considered. Therein, #G denotes the number of annotated notes. It shows that the most common errors from all algorithms were octave errors. Note segmentation has never been studied in the tremolo case; thus, here, we can only provide the F/P/R achieved using the method presented in Section 3.3. The method obtained F/P/R values of 0.8834/0.8455/0.9248, respectively.

#### 4.2.3. Running Speed

All the algorithms proposed in this paper were implemented in Matlab. Even without code optimization, the methods exhibited overwhelming speed advantages. Running on hardware with an AMD R7-5800H CPU, 32GB RAM, and Matlab 2023b with the correct boundaries, the zero-crossing-based pitch estimation method demonstrated an approximate 26 and 27 times increase in speed compared to the C++-implemented pYIN (pYIN code available: https://code.soundsoftware.ac.uk/projects/pyin, accessed on 17 February 2025) in the SPE and PES processes. Moreover, it was 108 and 65 times faster than the Matlab-implemented BNLS (BNLS code available: https://github.com/LimingShi/Bayesian-Pitch-Tracking-Using-Harmonic-model, accessed on 17 February 2025).

## 5. Discussion

The results show a significant improvement on the first three excerpts in the note onset detection task. The decrease in recalls in *Ambush from Ten Sides* was caused by the attack–transient intervals being shorter than 9.09 ms, which were produced by the high-speed tremolo as shown in Figure 3A. The extrema-with-padding method alleviated this issue at the expense of precision. Therefore, we recommend the extrema-with-padding method if the overall tremolo rate exceeds 15 Hz. The features in other onset detectors fully or partially consider the amplitude information; however, the zero-crossing and extrema time intervals around the arched forms are invariant to the amplitude and separated from the pitch period range. Therefore, the success of this new feature is partly attributed to the photoelectric sensors’ full-band response.

In the vibration signal, the extrema generally generate more candidates than zero-crossings, resulting in initial pitch selection being more difficult and a lower pitch quality (see Figure 6). Catastrophic results were recorded by the probabilistic YIN (pYIN) and Bayesian non-linear least square (BNLS) methods using the segmentation before the pitch estimation (SPE) process, as shown in Table 3. This implies that our methods effectively resist non-stationarity of short-time approaches, and pYIN and BNLS to some extent rely on the interclip pitch continuity to achieve better results.

Continuous time-period mapping (CTPM) provides a way to visualize the periodicity in the waveform and addresses the lack of pitch synchronism in the empirical mode decomposition (EMD) algorithm. Although the noise-sensitive drawback in EMD remains [23], the photoelectric pickups that directly capture the string displacement exclude external noise. In addition, compared to the harmonic spectrum and autocorrelation function (ACF)-like features, CTPM has a much lower computational consumption. In summary, an instrument with this type of low-cost automatic music transcription (AMT) module can be played live with ambient noise.

## 6. Conclusions and Future Work

In this paper, we propose high-resolution transcription algorithms for photoelectric signals from a pipa based on zero-crossing and extremum points. This method leverages the morphological advantages of photoelectric sensor signals over audio and electromagnetic signals during plucking. Additionally, a novel time–frequency feature is used to address waveforms with multiple oscillations in a period and pitch tracking. In this study, we benchmarked the performance of six short-time onset detection and pitch estimation algorithms for various playing techniques. The proposed zero-crossing- and extremum-based boundary detection methods achieved superior F-scores in most note onset cases while maintaining a high recall and resolution. In high-speed tremolo scenarios, padding the extrema significantly improved recall. For vibration signal pitch estimation, our method achieved a relatively high performance level and greater efficiency. In combination with post-processing, automatic music transcription (AMT) for mixed playing techniques was achieved using our process.

In the future, we will focus on improving the performance and running time of these algorithms. Furthermore, transcribed pitch curves can be used to analyze pitch-shift techniques and expand the system’s functionality. CTPM has potential for further exploration in other periodic signals and research fields. Finally, our method using photoelectric signals could be used on other plucked instruments such as the guitar in the future.

## Figures and Tables

**Figure 1 sensors-25-01361-f001:**
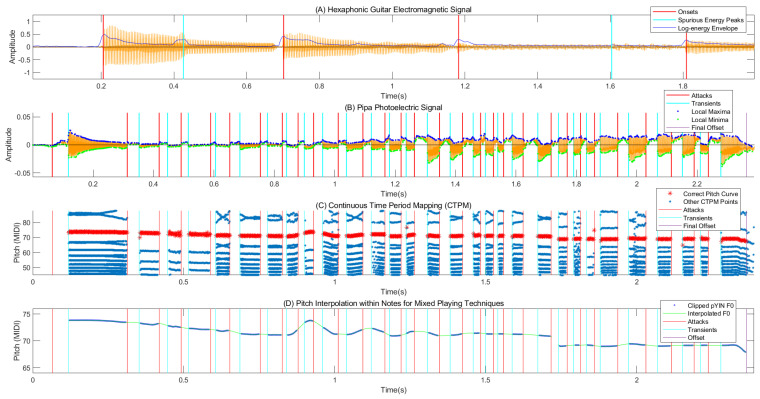
(**A**,**B**) Signals from the electromagnetic pickup on the guitar and the photoelectric pickup on the pipa. (**B**) Onset detection using extrema-based envelopes. (**C**) Continuous time-period mapping (CTPM, refer to Section 3.1 for more details) and pitch curves of the photoelectric signal from plot (**B**) that contains a mixture of tremolo and pitch-shift techniques. (**D**) Interpolation of the pitch curves to fill the gaps within tremolo notes, where probabilistic YIN (pYIN) [12] denotes a non-parametric pitch estimator used in this paper.

**Figure 2 sensors-25-01361-f002:**
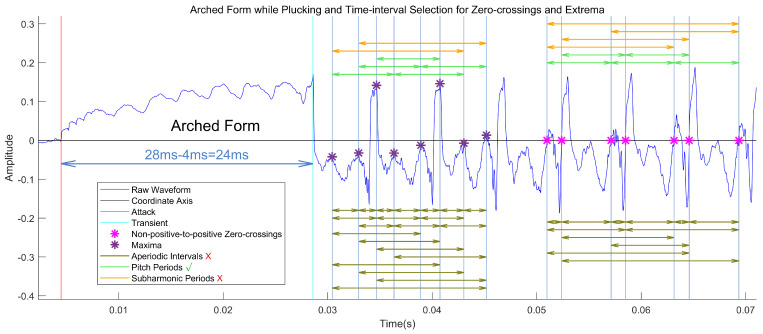
An example of an arched feature for onset detection and time-intervals based on non-positive-to-positive zero-crossings and maxima for pitch estimation. The interval between attack and transient is much greater than those of adjacent feature points after the transient, the neighboring pitch periods are approximately identical, and the subharmonic period is usually closer to an integer multiple than to the pitch period around.

**Figure 3 sensors-25-01361-f003:**
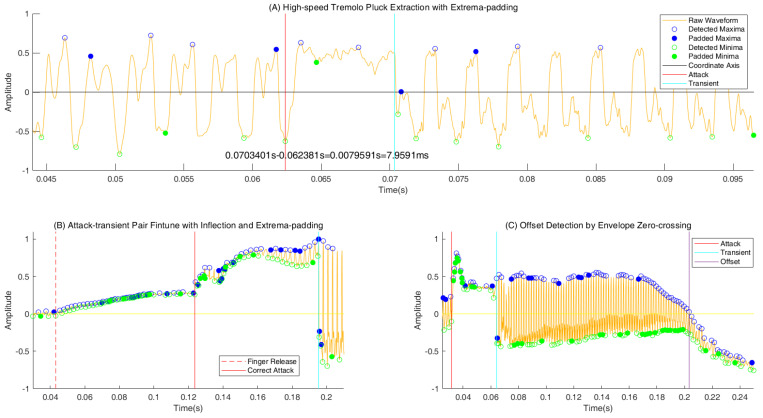
Specific examples of boundary detection.

**Figure 4 sensors-25-01361-f004:**
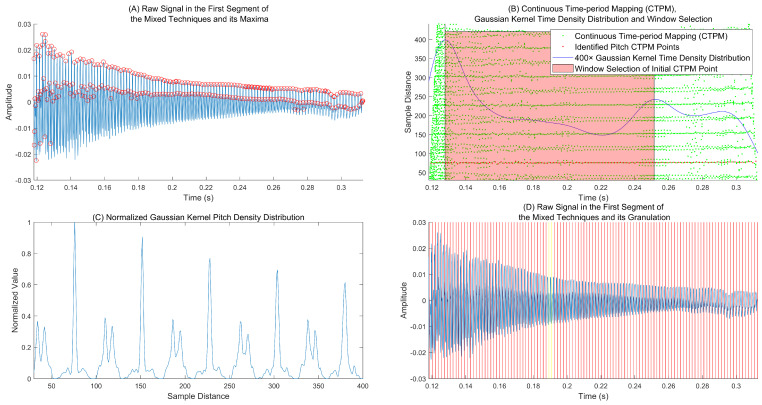
Specific process for pitch estimation: (**A**) The first clip signal of the mixed playing technique in Figure 1B and its maxima; (**B**) the computed continuous time-period mapping (CTPM), and the Gaussian kernel density function along the time axis; the pink region was used to estimate the initial period; (**C**) the Gaussian kernel density function through the sample distance axis of the CTPM points in the pink region; (**D**) forward-backward track of the pitch marker sequence; the yellow lines represent the extrema pair of the initial CTPM point.

**Figure 5 sensors-25-01361-f005:**
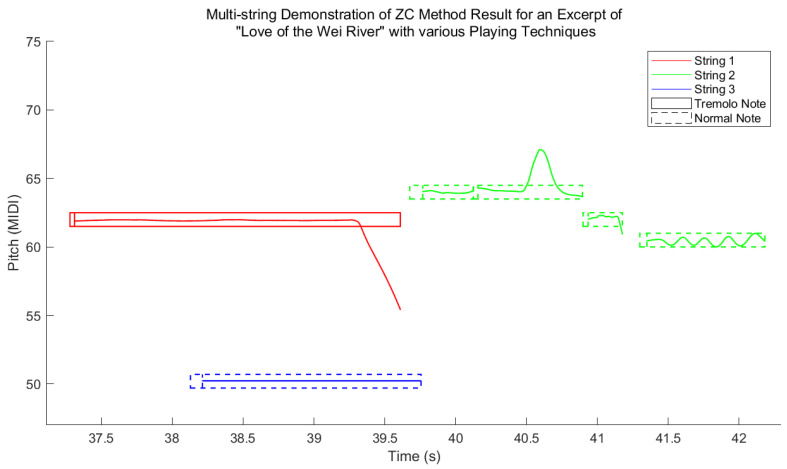
Final transcription features from an expert of *Love of the Wei River* with multiple playing techniques, where ZC denotes zero-crossing.

**Figure 6 sensors-25-01361-f006:**
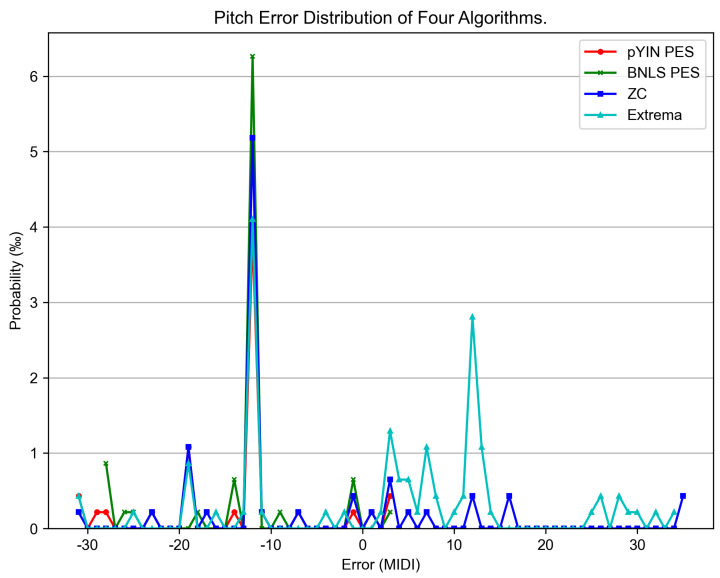
Global pitch error histogram, where pYIN PES, BNLS PES, ZC, and Extrema denote the probabilistic YIN [12] and Bayesian non-linear least square [16] methods under pitch estimation before the segmentation process, zero-crossing-based methods, and extrema-based methods, respectively.

**Table 1 sensors-25-01361-t001:** Average note onset detection performance and mean absolute error (MAE) of true positives to ground-truths. Bold numbers indicate the optimal value for each metric and each song.

Method	*Jasmine Flower*		*Nanni Bay*		*Love of the Wei River*		*Ambush from Ten Sides*	
	F/P/R (%)	MAE (ms)	F/P/R (%)	MAE (ms)	F/P/R (%)	MAE (ms)	F/P/R (%)	MAE (ms)
SpecFlux	80/91/72	21.9	84/83/86	21.3	30/22/46	19.6	57/60/55	18.3
SuperFlux	70/**96**/55	21.7	80/**88**/73	21.3	27/27/26	21.4	**60**/72/52	18.0
ComplexFlux	72/**96**/58	21.4	78/**88**/70	21.3	25/23/28	21.2	59/76/49	17.7
Log-Energy	80/71/92	2.3	58/42/93	2.9	22/13/**92**	3.0	47/43/54	**5.6**
ZC	88/85/92	**1.8**	**89**/87/92	**2.4**	**55**/**40**/90	4.0	52/**81**/38	6.5
Extrema-without-padding	**90**/85/**96**	2.4	**89**/84/94	2.5	54/39/88	**2.9**	51/69/40	6.8
Extrema-with-padding	83/73/95	2.5	81/70/**95**	2.7	42/27/**92**	3.7	54/43/**73**	7.0

**Table 2 sensors-25-01361-t002:** Tremolo note recognition (TNR), tremolo pluck detection (TPD) performance, and mean absolute error (MAE) of true positives to ground-truths. Bold numbers indicate the optimal value for each metric and each song.

Method	*Nanni Bay*			*Love of the Wei River*			*Ambush from Ten Sides*		
	TNR (%)	TPD (%)	MAE (ms)	TNR (%)	TPD (%)	MAE (ms)	TNR (%)	TPD(%)	MAE (ms)
SpecFlux	77/62/**100**	83/97/73	20.3	57/40/**100**	67/99/51	17.2	64/88/50	60/98/43	17.9
SuperFlux	87/77/**100**	69/97/53	20.1	67/50/**100**	45/**100**/29	17.8	55/95/39	49/98/33	17.8
ComplexFlux	91/83/**100**	62/97/46	19.5	67/50/**100**	46/99/30	17.4	55/90/39	43/98/28	17.7
Log-Energy	45/29/**100**	86/96/79	3.2	53/36/**100**	68/98/52	5.5	63/68/59	67/96/51	**6.9**
ZC	**100**/**100**/**100**	98/**100**/96	3.0	**89**/**80**/**100**	93/99/87	6.5	69/**100**/52	65/**99**/48	8.0
Extrema-without-padding	95/91/**100**	**99**/**100**/**99**	**2.7**	**89**/**80**/**100**	**98**/99/96	**4.7**	78/97/65	74/98/60	7.5
Extrema-with-padding	56/38/**100**	**99**/98/**99**	2.8	62/44/**100**	**98**/97/**100**	5.0	**82**/73/**93**	**89**/95/**85**	7.1

**Table 3 sensors-25-01361-t003:** Average intersection over union (IOU) to evaluate voice activity detection and the pitch estimation accuracy under PES and SPE processes. The voicing results of the extrema approaches with and without padding are both indicated.

Metrics	Methods			
	pYIN (%)	BNLS (%)	ZC (%)	Extrema (%)
VAD	42.6	34.7	**60.1**	51.3/50.5
PES	**90.9**	89	N/A	N/A
SPE	29.2	29.6	**87.7**	83.9

## Data Availability

The data presented in this study are openly available in https://github.com/veneno1213822/CTPM_DATA, accessed on 17 February 2025.

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
