# Peer review of "Towards Automatic Expressive Pipa Music Transcription Using Morphological Analysis of Photoelectric Signals"

_sensors, 2025, doi:10.3390/s25051361_

Round 1

Reviewer 1 Report

Comments and Suggestions for Authors

At first sight this paper seems to be quite interesting. The topic is timely, and the methodology is very sophisticated. After continued reading, however, there is a kind of frustration as the readability and understandability of the paper is quite challenging. The abundant use of abbreviations, technical terms without explanation and very dense and cryptic style of writing makes it very difficult to understand the content and—for reviewers—to evaluate the standards of the paper. It is my impression that this paper has a lot of potential, but it should be modified substantially to convert it into a readable paper that can be accepted for publication. Given that the paper is about music and a music instrument, it should be understandable also to some extent for music scholars with at least some background knowledge of digital instruments. The sophistication level of the current text, however, is beyond the level of common readers. I should recommend therefore to expand the paper a little, by adding some more intuitive descriptions and explanations so as to lower the threshold of comprehensibility. As a reviewer with a considerable background in mathematics and signal analysis, I found it very difficult to understand the second half of the paper. This should be even more so for readers without the needed background. I list below some general remarks and detailed comments with the aim to improve the readability of the paper.

General remarks

-      The contents of the paper are very interesting and timely.

-      The English language use is fluent and idiomatic.

-      There are too many comments about the structure of the paper and not enough comments on the technical stuff, which needs more intuitive descriptions.

-      The abundant use of abbreviations, with some of them even not explained in full, makes reading very hard. It makes a lot of sense to provide the abbreviations, especially for use in figures and tables, but using abbreviations instead of full text makes reading quite difficult. Here is a list of most of them: SMI, AMT, APT, CTPM, UVD, VAD, NLS, BNLS, ACF, RMS, ZC, EMD, IMF, DIO, PSOLA, PES, SPE, pYIN, ... Which reader is able to retain all of these after presenting them only once? Reading then becomes a process of “studying” rather than “reading”, and the feeling of the reader is one of frustration. It gives the reader also a feeling of pedantry. Abbreviations should not be a goal by themselves, and using the full spelling makes reading much easier.

-      The used methodology and measurement tools are quite sophisticated and seem to have a lot of potential. It is important, however, to explain them also somewhat more intuitively before going in the technicalities of description. The reader without the needed background knowledge in signal analysis should be able also to have a glimpse of what is meant. Expanding a little here can improve the text considerably.

-      The figures are very beautiful and well-organized, but more accompanying text is needed to explain how to read and interpret them. Each term in the figures should be explained at least a little in intuitive terms. This can be very short, but it is very important to make the paper somewhat more self-sufficient, which means that it should be possible to read the paper without need of consulting the added references. This is not easy, but doing this is the hallmark of good writers.

Detailed comments

-      Lines 20 and 27: The introduction of the abbreviation (SMI, AMT) makes a lot of sense here. This combination of abbreviation and writing out in full between brackets should be maintained throughout the whole paper, but in case of too many abbreviations, it makes reading easier when the full text is provided in the main text (see general comments).

-      Line 48: here is an example of raising the level of readability. Compare the more expanded phrasing: Automatic pipa transcription (APT), as an automatic music transcription dedicated to, ...

-      Line 69: explain very shortly why you use the exponent (n-69)

-      Line 81: what is meant with an “arched figure”. Explain shortly in intuitive terms.

-      Figure1: this is a very beautiful figure. Some more explanation is needed, however, to understand it. This holds in particular for (C): the continuous time period mapping. It is not clear how to interpret this figure. Please explain sufficiently in detail.

-      Line 106: try to keep the comments on the structure of the paper more limited.

-      Lines 111 ff: here the abbreviations become quite challenging and confounding. It still makes sense to introduce them in brackets at first appearance, but using them consistently through the paper makes reading very hard. Readers may start to become frustrated here and perhaps decide to stop reading.

-      Line 124; what does pYIN mean? Please explain.

-      Line 132: please provide some minimal intuitive explanation about SpecFlow, SuperFlux and ComplexFlux. What is the aim of these algorithms and what do they actually do?

-      Line 136: same remark about the zero-crossing. Even is this may a common concept for man? readers, it should be explained why this measurement tool is used, and which information it provides. Again, provide an intuitive description of this conceptual tool. This can even be very short, but it helps a lot to improve the readability of the description.

-      Line 150: what is the meaning of PM candidates? And what is the additional value of using an abbreviation here? It only makes reading more difficult.

-      Line 150: what is a DIO algorithm? Explain shortly.

-      Lines 157-158: very technical sentence. Please explain the used terms more intuitively.

-      Line 160: the following “article”. What follows is not a next article, but mere text.

-      Figure 2 must be explained more in detail. Expand a little on the concepts of boundary and pitch extraction.

-      Figure 4: provide CTPM in full and the abbreviation between brackets in the figure caption. Provide also some more intuitive description in the main text of the concept of Gaussian kernel density. Line 202: the concept of padding may be very new for common readers. Please explain shortly at first appearance of the term.

-      Line 221: extrema instead of extrama?

-      Lines 229-lines 374: I stopped reading here, as I felt not enough qualified to evaluate the content.

-      Line 376: zero-crossing and extrema points. These concepts seem to be quite important. It makes sense therefore to expand a little at their first introduction in the text to explain in more intuitive terms what they mean and what they are used to. This hold also for the term morphological, which appears also in the title, but which should be discussed also in the introduction. It is important that the reader has a kind of take-home message after reading, and the morphological aspect should be one of them. The core ideas of the title must be explained also intuitive terms.

Author Response

Comments 1: Zero-crossings and extrema and arched figure (it may refer to the arched feature).

Response 1: Thank you for pointing these out. We have added a new paragraph including the formulation about modulated periodic signal, description about zero-crossing/extrema, and time interval selection problem (lines 86-101) with a new figure (current figure 2) in Introduction.

Comment 2: Abbreviation issues.

Response 2: Thank you for pointing these out. For SMI, AMT, APT, CTPM, UVD, VAD, NLS, ACF, RMS, EMD, IMF, PSOLA, we have kept the full name+abbreviation combination in whole paper. For CTPM, pYIN, BNLS, PES, SPE, we have only added the full name at the beginning of a section or a figure due to numerous occurrences.We have removed the abbreviations ZC and PM, except the first presence and those in the figures.

Comment 3: More explanation of pYIN and DIO, SpecFlow, SuperFlux and ComplexFlux, granular synthesis, PSOLA, padding.

Response 3: Thank you for pointing these out, we have clarified these concepts with minimal words and mathematical formulas.

Comment 4: MIDI number exponent n-69

Response 4: Thanks for pointing these out, exponent n-69 is involved with the definition of MIDI-to-frequency conversion, and the coefficient 69 has a coincidence with the line number in old version. We have added a reference for this definition.

Reviewer 2 Report

Comments and Suggestions for Authors

The manuscript is generally well-organised. The technical content of the paper is generally fine. The major issue of the manuscript is the writing skill. 

First and most importantly, the manuscript should put itself in a clear shape. The manuscript is a paper about signal processing techniques. As a result,  mathematical presentations are expected. Figure 2 seems misleading to the readers as such a diagram is usually shown in a machine learning-related paper. 

Secondly, the author puts the word "expressive" in the title, but such a term is barely seen in the manuscript. The author should justify how the proposed method upholds a connection with the term "expressive". For example, does the proposed method use expressiveness information to help signal processing, or could the proposed signal processing method be used for expressive performance? 

Thirdly, the presentation of the novelty of the paper is poor. The author should raise a research problem first and then present how the proposed paper can solve such problems. Moreover, the result should support a scientific hypothesis to help the reader understand the proposed problems better. 

Finally, the manuscript should justify how the proposed measures properly evaluate the proposed solution. 

Author Response

Comments 1: The manuscript is a paper about signal processing techniques. As a result,  mathematical presentations are expected. Figure 2 seems misleading to the readers as such a diagram is usually shown in a machine learning-related paper. 

Response 1: Thank you for pointing this out. We have added mathematical presentations for modulated period signal, pYIN, specflux, superflux, complexflux and metrics like F/P/R IoU etc. As an interdisciplinary paper, we insist on retaining Figure 2 (current Figure 3), as we hope this article could serve as a reference for the future AMT module design in SMI, removal of this figure may cause confusion in the background and specific work in this paper.

Comments 2:  the author puts the word "expressive" in the title, but such a term is barely seen in the manuscript. The author should justify how the proposed method upholds a connection with the term "expressive". For example, does the proposed method use expressiveness information to help signal processing, or could the proposed signal processing method be used for expressive performance? 

Response 2: Thank you for pointing this out. We have added a sentence to clarify this term. [Unlike prototypical performance without variation of auditory parameters, expressive performance is carried out by various playing techniques that often accompany intensive strength and pitch variations, which will be addressed by the proposed methods in paper.]

Comment 3: the presentation of the novelty of the paper is poor. The author should raise a research problem first and then present how the proposed paper can solve such problems. Moreover, the result should support a scientific hypothesis to help the reader understand the proposed problems better. 

Response 3: Thank you for pointing this out. We have added a new paragraph including the formulation about modulated periodic signal, clarification about zero-crossing/extrema, and time interval selection problem with a new figure in Introduction section (lines 86-101 and current Figure 2).

Comment 4: the manuscript should justify how the proposed measures properly evaluate the proposed solution.

Response 4: Thank you for pointing this out. We have reorganized the description of 2 tremolo measures and added more detail about pitch estimation accuracy which are highlighted in lines 324-348.

Round 2

Reviewer 2 Report

Comments and Suggestions for Authors

Three major issues:

1. why does the author use "Smart-Pipa" in the title? This term is again misleading. The manuscript is about signal processing, not instrument design. The current title suggests a new instrument (potentially an electronic one) is proposed. The author must change the title. 

2. Regarding Fig. 3, the diagram makes little sense. The caption could provide further details. The author must re-design the diagram as the diagram contains little helpful information. 

3. The scientific hypothesis is still missing, although the presentation of results sounds better now. By comparing the different types of features, why does a specific feature outperform other features? The author must give a reason for the result. Usually, the justification should be due to a unique property of Pipa (timbre of Pipa) or a specific feature that would work better. 

Finally, automatic music transcription is a typical signal-processing task. Being able to transcribe an expressive performance is a significant achievement. The author should not attempt to establish a connection between the proposed algorithm and any AI concepts, even in this AI era. 

Author Response

Comments 1: why does the author use "Smart-Pipa" in the title? This term is again misleading. The manuscript is about signal processing, not instrument design. The current title suggests a new instrument (potentially an electronic one) is proposed. The author must change the title.

Response 1: Thank you for pointing these out. We have deleted the “smart pipa” in the title and its related words.

Comment 2: Regarding Fig. 3, the diagram makes little sense. The caption could provide further details. The author must re-design the diagram as the diagram contains little helpful information.

Response 2: Thank you for pointing this out. We have deleted the Fig. 3 and its related words.

Comment 3: The scientific hypothesis is still missing, although the presentation of results sounds better now. By comparing the different types of features, why does a specific feature outperform other features? The author must give a reason for the result. Usually, the justification should be due to a unique property of Pipa (timbre of Pipa) or a specific feature that would work better.

Response 3: Thank you for pointing these out, We have added the connection among the pitch range, the sensors and the algorithms to clarify the engineering innovation and the basic assumption of the algorithms. In Introduction Section:

[Furthermore, the time intervals in our proposed methods are mainly based on the pipa register ranging from 110 to 1318.5 Hz, low frequency information (<110 Hz)  of the raw signal is used for onset detection, and high frequency information (>1318.5 Hz) must be neglected in all of our proposed methods. The arched form, which benefits from the full-band response of the photoelectric sensors, can be seen as a novel low-frequency feature of plucked instruments.]

We have modified lines 77-79 by:

[The arched form, which is not displayed in the signal from microphone and electromagnetic pickups as a result of their suppressed low-frequency response, could circumvent the amplitude fluctuation introduced by pitch-shift techniques such as vibrato and sliding during signal vibration.]

We have added an description for result in Discussion Section:

[Unlike the features in other onset detectors which fully or partially consider the amplitude information, the time intervals of zero-crossings and extrema around the arched forms are invariant to the amplitude and separated from the pitch period range. Therefore, the success of this new feature is partly attributed to the full-band response of the photoelectric sensors.]

We have added a new future work in Conclusion Section to expand its application scope:

[Such a paradigm using the photoelectric signal could be shifted to other plucked instruments such as guitar in the future.]